# Fungal–bacteria interactions provide shelter for bacteria in Caesarean section scar diverticulum

**Peigen Chen**[1,2,3†], **Haicheng Chen**[1,2,3†], **Ziyu Liu**[1,2,3], **Xinyi Pan**[1,2,3], **Qianru Liu**[1,2,3], **Xing Yang**[1,2,3]*

[1]Reproductive Medicine Center, The Sixth Affiliated Hospital, Sun Yat-sen University, Guangzhou, China; [2]GuangDong Engineering Technology Research Center of Fertility Preservation, Guangzhou, China; [3]Biomedical Innovation Center, The Sixth Affiliated Hospital, Sun Yat-sen University, Guangzhou, China

*For correspondence:
yangx52@mail.sysu.edu.cn

†These authors contributed equally to this work

Competing interest: The authors declare that no competing interests exist.

**Abstract** Caesarean section scar diverticulum (CSD) is a significant cause of infertility among women who have previously had a Caesarean section, primarily due to persistent inflammatory exudation associated with this condition. Even though abnormal bacterial composition is identified as a critical factor leading to this chronic inflammation, clinical data suggest that a long-term cure is often unattainable with antibiotic treatment alone. In our study, we employed metagenomic analysis and mass spectrometry techniques to investigate the fungal composition in CSD and its interaction with bacteria. We discovered that local fungal abnormalities in CSD can disrupt the stability of the bacterial population and the entire microbial community by altering bacterial abundance via specific metabolites. For instance, *Lachnellula suecica* reduces the abundance of several *Lactobacillus* spp., such as *Lactobacillus jensenii*, by diminishing the production of metabolites like *Goyaglycoside A* and *Janthitrem E*. Concurrently, *Clavispora lusitaniae* and *Ophiocordyceps australis* can synergistically impact the abundance of *Lactobacillus* spp. by modulating metabolite abundance. Our findings underscore that abnormal fungal composition and activity are key drivers of local bacterial dysbiosis in CSD.

## eLife assessment

This **important** study reports the fungal composition and its interaction with bacteria in the Caesarean section scar diverticulum. The data are **solid** and supportive of the conclusion. This work will be of interest to researchers and clinicians who work on women's health.

## Introduction

Caesarean section (CS) is a prevalent surgery worldwide, and its rate has increased in recent decades (*Verguet et al., 2015*). Although CS can significantly reduce dystocia and stillbirths (*Keag et al., 2018*), Caesarean section scar diverticulum (CSD) affect about 19.4–88% of women receiving this operation (*Tower and Frishman, 2013*). CSD emerges from poor healing of the local uterine incision, forming a depression or cavity that connects with the uterine cavity (*Poidevin, 1959*). Recently, CSD has drawn widespread attention because of its potential damage to subsequent fertility. For example, Gurol-Urganci et al. found that the possibility of subsequent pregnancy decreases by an average of 10% after CS relative to a previous vaginal delivery (*Gurol-Urganci et al., 2013*). A niche can reduce the chances of embryo implantation and increase the likelihood of spontaneous miscarriages if the

implantation occurs close to or in the CSD (*Naji et al., 2013*). Our previous studies have shown that the persistent effusion of CSD is a key cause of failed embryo implantation (*Cai et al., 2022*).

The unique microbial community composition in the female reproductive tract is essential in maintaining female reproductive health (*Evans et al., 2016*; *Benner et al., 2018*; *Koedooder et al., 2019*). Our previous studies have demonstrated that the abnormal alterations in the local microbiota of Caesarean section scar diverticulum (CSD) cause continuous leakage through local inflammation and immune imbalance (*Yang et al., 2021*). Further investigations of ours revealed the underlying mechanism demonstrating that abnormal bacteria in CSD deplete protective fatty acids and generate *N*-(3-hydroxy-eicosanoyl)-homoserine lactone, thus leading to promoting apoptosis of vascular endothelial cells and endometrial epithelial cells, ultimately impairing women's reproductive capacity (*Yang et al., 2022*). Nonetheless, the causative factors behind bacterial dysbiosis in complex microbial communities remain poorly understood.

Fungi and bacteria, as integral components of the human microbiome, establish complex interactions with each other and the host. The collection of genomes and genes carried by fungal species coexisting within a specific environmental or biological niche is commonly called as the 'mycobiome' (*Ghannoum et al., 2010*). However, in CSD, the mycobiome remains poorly understood, particularly in comparison to the microbiome. The interaction between microbial communities in the reproductive tract, particularly bacteria and fungi, is critical in maintaining female reproductive health (*Bradford and Ravel, 2017*). Therefore, this study aims to examine the composition and function of fungi in CSD, as well as their interactions with bacteria, to provide a more comprehensive understanding of the role of microbial communities in CSD. This study also aims to identify potential therapeutic approaches to enhance fertility by building on current research.

## Results

### Recruitment of participants and metagenomic sequencing

Forty-eight participants were included in this study, including 24 in the CSD group and 24 in the CON group. The clinical characteristics of the participants are shown in *Table 1*. To better characterize the composition of the cervical microbiota, we performed ultra-high-depth metagenomic sequencing. The effective data volume of each sample ranged from 14.25 to 18.63 G, the N50 statistical distribution of contigs ranged from 230 to 245 bp, and the number of open reading frames (ORFs) in the gene catalogue (non-redundant gene set) constructed after redundancy removal was 281,107.

### Composition and characteristics of bacterial communities

Alpha diversity based on Pielou index (*Figure 1A*) and Shannon index (*Figure 1B*) calculations showed that species richness and evenness in the CSD group were significantly higher than those in the control group (*T*-test). Beta diversity based on the Bray–Curtis distance (*Figure 1C*) indicated that the distances between samples within the CSD group were greater than those in the CON group (PERMANOVA test).

The composition of community state types (CST) in the CSD group and CON group differed. The proportion of CST III decreased in the CSD group, while the more unstable CST IV-B and CST IV-C increased (*Figure 1D*). The CST community types in the CON group were dominated by *Lactobacillus* spp. (*Figure 1D*). CST subtype analysis revealed more detailed community types. In CST I (dominated

**Table 1.** Clinical characteristics of the participants included in the study.

|  | CON group (*n* = 24) | CSD group (*n* = 24) | p value |
| --- | --- | --- | --- |
| AMH (ng/ml) (mean (SD)) | 3.69 (2.09) | 3.00 (2.32) | 0.288 |
| BMI (mean (SD)) | 21.04 (2.25) | 22.25 (2.70) | 0.1 |
| Basal FSH (IU/l) (mean (SD)) | 6.99 (1.42) | 7.70 (1.88) | 0.144 |
| Basal LH (IU/l) (mean (SD)) | 5.51 (2.97) | 5.91 (3.55) | 0.676 |
| Basal E2 (pg/ml) (mean (SD)) | 47.42 (68.52) | 44.49 (29.46) | 0.848 |
| infertile years (mean (SD)) | 3.54 (2.19) | 4.54 (2.90) | 0.184 |

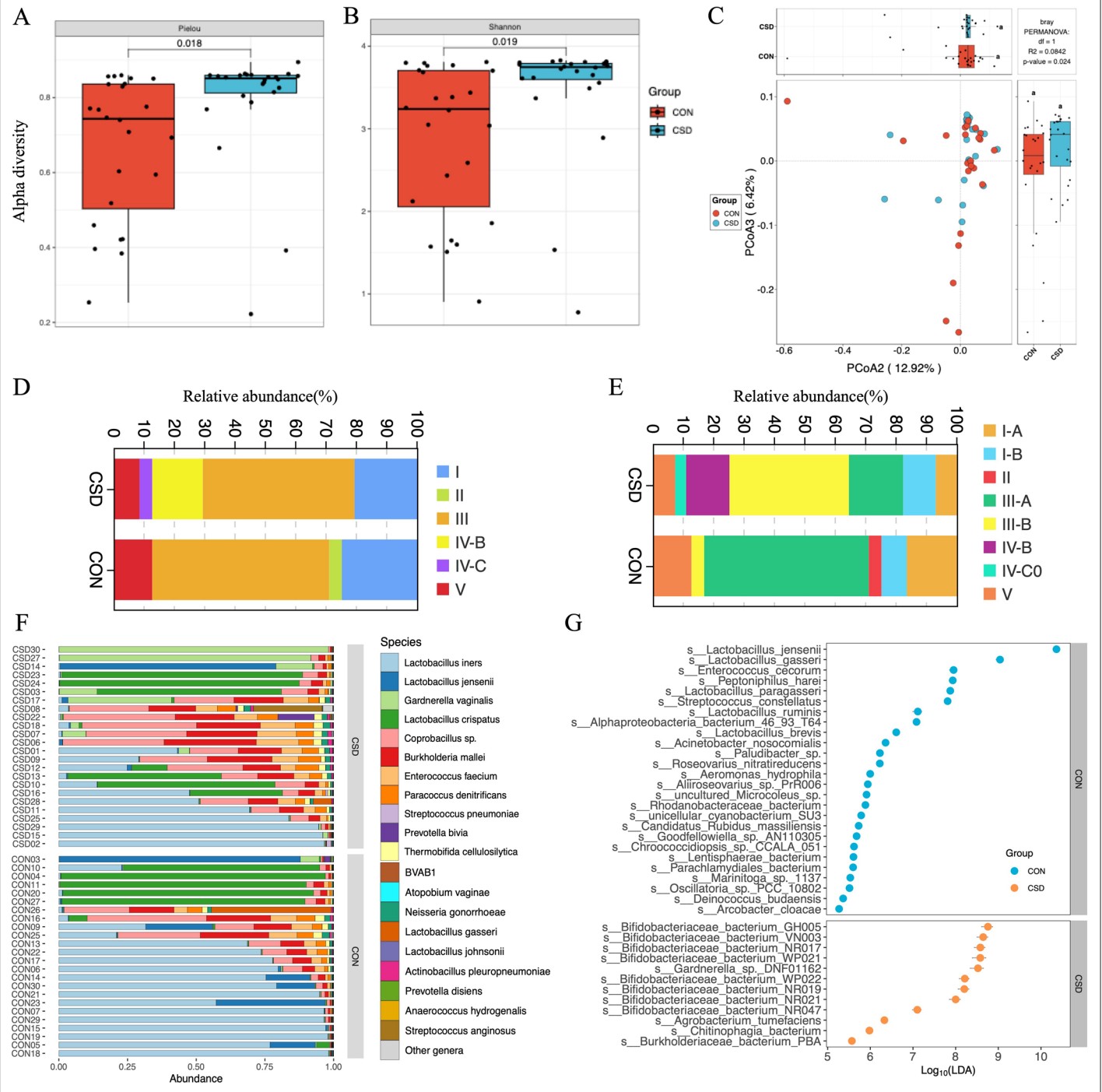

**Figure 1.** Bacterial community structure and composition characteristics. Alpha diversity based on Pielou index (**A**) and Shannon index (**B**); (**C**) beta diversity based on the Bray–Curtis distance; stacked graph of community state type (CST) (**D**) and subCST (**E**) compositions; (**F**) stacked plot of species composition; (**G**) scatter plot of differentially abundant species between CSD and CON groups.

by *L. crispatus*) and CST III (dominated by *L. iners*), the CSD group was mainly distributed in the B subtypes with a lower abundance of *L. crispatus* and *L. iners* (*Figure 1E*). The stacked bar chart of species composition shows the species composition characteristics of each sample (*Figure 1F*). One hundred and sixty-two differential species were identified between the two groups. The differential species analysis indicated that *Bifidobacteriaceae bacterium* spp. and *Gardnerella* spp. were significantly higher in the CSD group than in the CON group (*Figure 1G*).

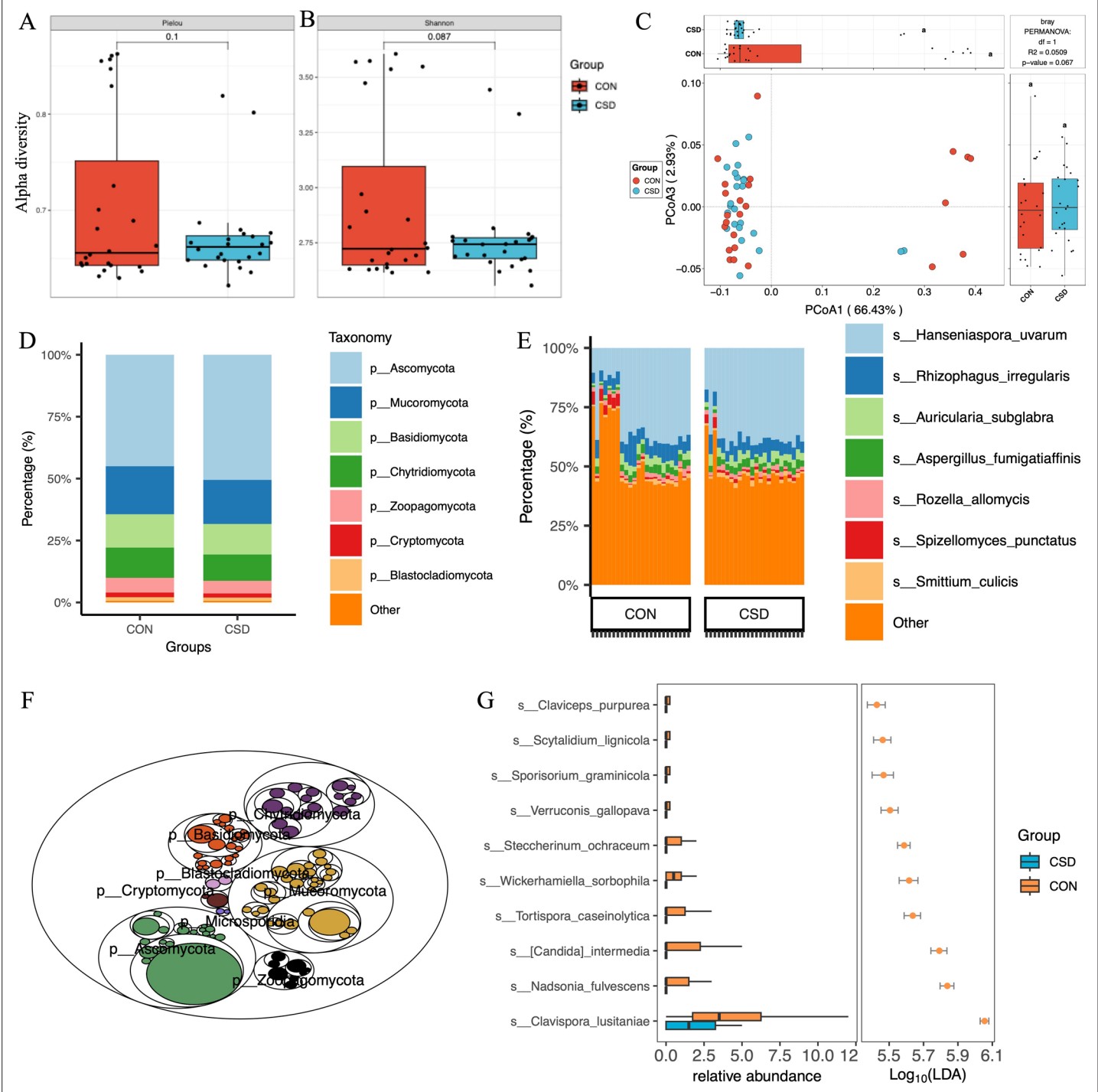

**Figure 2.** Fungal community structure and composition characteristics. Alpha diversity based on Pielou index (**A**) and Shannon index (**B**); (**C**) beta diversity based on the Bray–Curtis distance; stacked graph of phylum level (**D**) and species level (**E**) compositions; (**F**) the tree map of fungal community; (**G**) box plot of differentially abundant species between CSD and CON groups.

The online version of this article includes the following figure supplement(s) for figure 2:

**Figure supplement 1.** Functional composition of the cervical microbiota.

## Composition and characteristics of fungal communities

A total of 431 fungal species were annotated (*Supplementary file 1*). There was no difference in α diversity based on the Pielou index (*Figure 2A*) and Shannon index (*Figure 2B*) between the two fungal communities. β diversity suggested similar distances between samples within each group (*Figure 2C*). Regarding species composition, the composition of the two groups was similar at the phylum (*Figure 2D*) and species (*Figure 2E*) levels. The tree map revealed the relationships among fungal phylum (*Figure 2F*). Forty-two differential species were identified between the two groups, and the top 10 species are shown in *Figure 2G*.

## The co-occurrence network among microbial community

We constructed co-occurrence networks for fungal species in the CSD and CON groups (*Figure 3A*). The network in the CSD group had fewer connections and weaker random robustness than the CON group (*Figure 3B*), indicating greater fragility of the CSD fungal co-occurrence network. Subsequently, a cross-domain network between bacteria and fungi was constructed (*Figure 3C*). The connection between bacteria and fungi in the CSD group was reduced compared with that in the CON group. At the same time, the CSD group had weaker random robustness than the CON group (*Figure 3D*). We selected species with a sample coverage rate greater than or equal to 60% for Spearman correlation analysis and visualized the data using heatmaps (*Figure 3E*) and networks (*Figure 3F*). The results showed that *Lachnellula suecica, Arthrobotrys oligospora*, and *Piptocephalis cylindrospora* have close relationships with bacteria (*Figure 3F*). *P. cylindrospora* has a close symbiotic relationship with *Lactobacillus jensenii* (*Figure 3F*).

## Functional gene analysis

Functional gene analysis indicates significant differences in the functional gene composition of microbial communities between the CSD and CON groups (*Figure 2—figure supplement 1A*). The CSD group has two significantly higher KEGG modules than the CON group, including M00142 (NADH: ubiquinone oxidoreductase, mitochondrion) and M00151 (cytochrome bc1 complex respiratory unit) (*Figure 2—figure supplement 1B*). The CSD group has significantly increased activity in the inflammatory mediator regulation of TRP channels and platelet activation, while the activity of the galactose metabolic process has decreased (*Figure 2—figure supplement 1C*). The CAZy annotation results indicate that the activity of GH73 (glycoside hydrolase family 73) in the CSD group has decreased (*Figure 2—figure supplement 1D*).

## Untargeted metabolomics revealed unique metabolic characteristics

Liquid chromatography/mass spectrometry (LC/MS) untargeted metabolomics was performed on 40 of the subjects. The heatmap shows differences in the distribution of metabolites between the two groups (*Figure 4A*). Enrichment analysis of signaling pathways suggests that the CSD group has significantly increased activity in the Thermogenesis and Pentose phosphate signaling pathways, while the activity in Steroid biosynthesis is significantly decreased (*Figure 4B*). Spearman correlation analysis suggests a close correlation between changes in the metabolite contents of *L. suecica* and *A. oligospora* (*Figure 4C*). At the same time, we also explored the relationship between bacterial and metabolite content changes (*Figure 4—figure supplement 1*).

Integrating bacterial, fungal, and metabolite data, we found that the fungus *L. suecica* has a mutual regulation relationship with *Lactobacillus* spp. through Goyaglycoside A and Janthitrem E (*Figure 5*). At the same time, *L. suecica* occupies a central position in the entire regulatory network and is closely related to changes in metabolites and bacterial composition. *Clavispora lusitaniae* and *Ophiocordyceps australis* are also fungal species that are closely related to changes in the abundance of *Lactobacillus* spp. (*Figure 5*).

## Discussion

CSD significantly affects female fertility, posing a challenge to women who wish to conceive. Our research team conducted a preliminary investigation on bacteria–host interactions (*Yang et al., 2022*). Current findings suggest that fungi play crucial roles in maintaining bacterial community and microbiome stability (*Zeise et al., 2021*; *Pareek et al., 2019*). This study explores the intricate

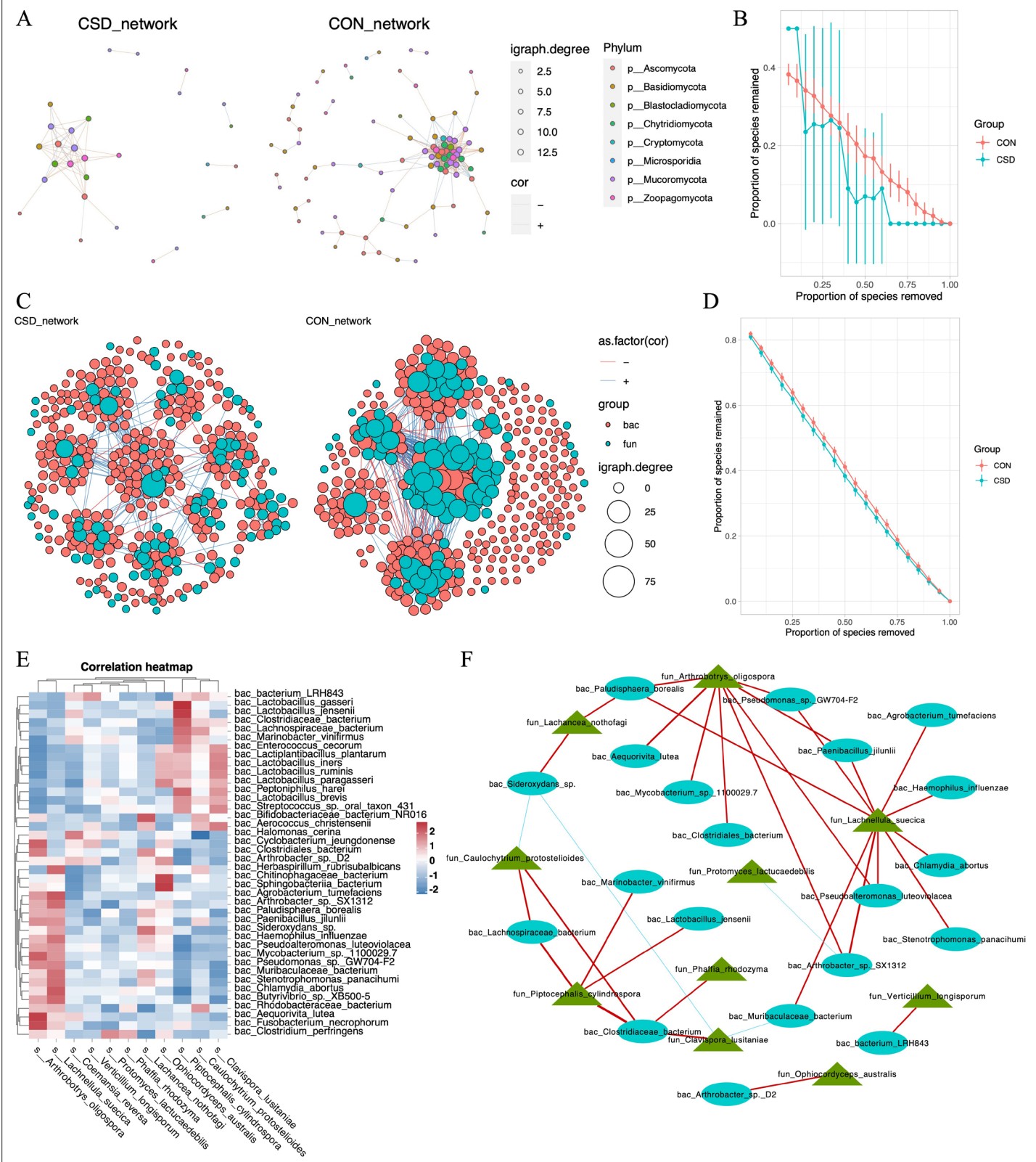

**Figure 3.** Interaction between bacteria and fungi. (**A**) Co-occurrence network of fungal communities; (**B**) comparison of the random robustness of co-occurrence network of fungal communities between CSD and CON groups; (**C**) interaction network between bacteria and fungi; (**D**) comparison of the random robustness of bacterial–fungal interaction network between CSD and CON groups; (**E**) heatmap of the correlation between bacteria and fungi; (**F**) network of correlations between bacteria and fungi ($R \geq 0.6$, $p \leq 0.05$).

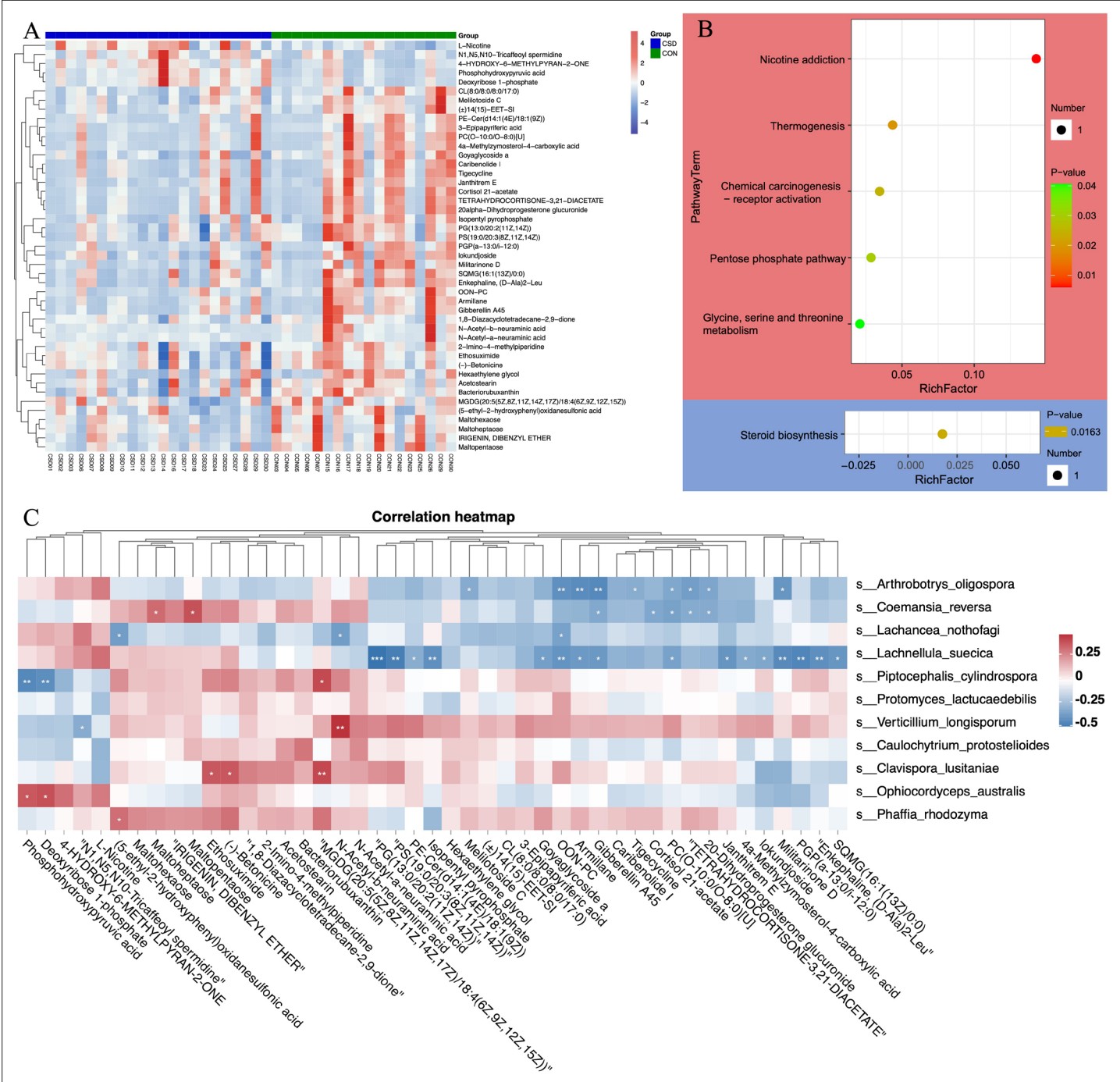

**Figure 4.** Metabolite composition and functional characteristics. (**A**) Heatmap of differentially abundant metabolites between CSD and CON groups. (**B**) Dot plot of KEGG enrichment analysis of metabolites. Red represents upregulation, blue represents downregulation. (**C**) Heatmap of the correlation between fungi and metaboliteds.

The online version of this article includes the following figure supplement(s) for figure 4:

**Figure supplement 1.** Heatmap of the correlation between bacterial and metabolites.

fungal–bacterial interactions within the microbiome of CSD, building on our team's previous research findings.

We achieved more precise bacterial community compositions by implementing higher-resolution metagenomics than our previous study (*Yang et al., 2022*). *L. jensenii* of the *Lactobacillus* genus exhibited the greatest discrepancy between the two groups. This species has been linked to integral

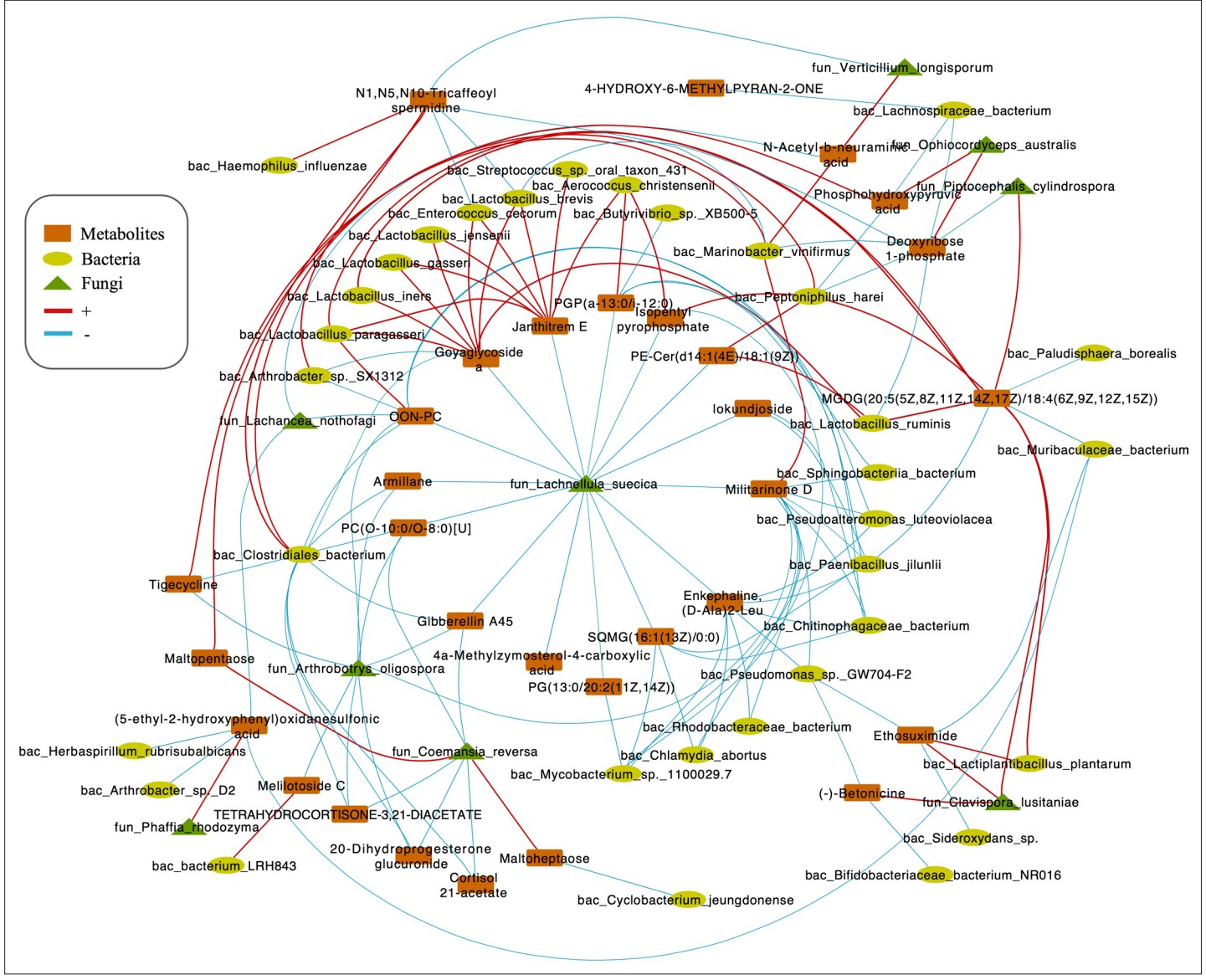

**Figure 5.** Correlations network of fungi, bacteria, and metabolites. The orange module represents metabolites. The yellow module represents bacteria. The green module represents fungi. The red line represents positive correlations. Blue line represents negative correlations. The thickness of the lines represents the magnitude of the correlation coefficient.

aspects of women's reproductive health. It has been demonstrated that a reduction in *L. jensenii* abundance has a close correlation with early-stage embryo arrest (***Wang et al., 2023***). Furthermore, research suggests that *L. jensenii* may hold potential in facilitating vertical transmission from mother to infant (***McCauley et al., 2022***).

Fungi and bacteria, as integral components of the human microbiome, establish complex interactions with each other and the host. The characteristics of fungal communities in CSD an CON group are similar, but there are differences in species composition. *Clabispora lusitaniae* is the species with the largest difference between the two groups. In addition, the co-occurrence network of fungal communities and the stability of the bacterial–fungal cross-domain network in the CSD group are poorer than those in the CON group, indicating that the cervical microbiota stability is disrupted in CSD, resulting in local microbial-immune imbalance. A previous study confirmed that persistent exudation in CSD is a crucial factor affecting embryo implantation (***Cai et al., 2022***). Several studies have shown that microbial dysbiosis can affect local immune balance and cause inflammation (***Hooper et al., 2012***; ***Littman and Pamer, 2011***).

Fungi play an important role in maintaining the stability of bacteria and the entire microbiome. Studies have shown that adding specific fungi can promote the growth of certain bacteria (*Pareek et al., 2019*), while fungi also play an important role in improving the adaptability of bacteria (*Pierce et al., 2021*). To further understand how abnormal fungal community status in CSD affects bacteria, leading to the disruption of local stability, we used LC/MS technique for metabolite detection. The metabolite analysis results showed differences in the metabolite spectra between the CSD and CON groups, and the regulatory mode of several different fungal species on metabolites was mainly negative regulation. By integrating fungal, bacterial, and metabolite information, we found that *L. suecica* played an important role in regulating the abundance of several *Lactobacillus* species. *L. suecica* reduces the abundance of several *Lactobacillus* species, including *L. jensenii*, by decreasing the metabolites Goyaglycoside A and Janthitrem E. Meanwhile, *C. lusitaniae* and *O. australis* can also synergistically affect the abundance of *Lactobacillus* by influencing metabolite abundance.

So far, we have conducted a comprehensive analysis of bacterial and fungal communities and changes in environmental metabolites in CSD. While our study provides significant insights, it is important to note that our methodology did not include the use of negative controls. This is a notable limitation, as negative controls are critical in distinguishing true microbial signals from potential contamination, particularly for rare species. The absence of negative controls could potentially affect the discrimination and identification of such rare species within our samples. Consequently, while our results provide valuable information on the prevalent microbial interactions within CSD, they should be interpreted with caution, especially when considering the implications for clinical treatment plans.

Furthermore, our analysis did not extend to examining the antibiotic sensitivity of the key fungi and bacteria identified, which limits the direct application of our results to clinical treatments. Future studies should consider including negative controls and antibiotic sensitivity analysis to validate our findings and extend the implications for clinical practice.

Despite these limitations, the study's results highlight the significant role of fungal composition and activity in driving bacterial dysbiosis in CSD. This insight is crucial for informing clinical treatment strategies and suggests that addressing fungal populations could be an important aspect of managing CSD. Future research is needed to further elucidate the complex interactions within the CSD microbiome and to develop more comprehensive treatment approaches that address both bacterial and fungal components.

## Materials and methods

### Participant recruitment and sample collection

The subjects were recruited at the Reproductive Medical Center of the Sixth Affiliated Hospital of Sun Yat-Sen University. The inclusion and exclusion criteria were previously described in a previous study (*Yang et al., 2022*). All participants were fully informed, volunteered to participate in this study, and signed informed consent forms. All study processes were reviewed and approved by the Sixth Affiliated Hospital Ethics Committee of Sun Yat-Sen University (IRB no. 2019ZSLYEC-005S).

The sample collection procedure is as described above (*Yang et al., 2022*). In short, after cleaning the external genitalia and vagina, a sterile disposable swab is inserted into the cervical canal, and the swab is rotated five times to collect the sample fully. The sample is rapidly transferred to liquid nitrogen for quenching and then stored at −80°C.

### DNA extraction and metagenomic sequencing

According to the manufacturer's instructions, the total DNA was isolated from the sample using QIAamp Fast DNA Mini Kit (QIAGEN, Hilden, Germany). The DNA concentration and integrity were evaluated by NanoDrop2000 spectrophotometer (Thermo Fisher Scientific, Waltham, MA, USA) and agarose gel electrophoresis. The DNA was fragmented with S220 Focused-ultrasonicators (Covaris, USA) and purified with Agencourt AMPure XP magnetic beads (Beckman Coulter Co, USA). Then, the library was constructed using the TruSeq Nano DNA LT Sample Preparation Kit (Illumina, San Diego, CA, USA) according to the manufacturer's instructions. The sequencing was performed using Illumina NovaSeq 6000.

## Raw data preprocessing

Parallel computing in research is implemented with GNU Parallel (*Tange, 2011*).

The sequencing raw data were stored in FASTQ files. After trimming and filtering with Trimmomatic (v0.36) (*Bolger et al., 2014*), the clean paired-end reads were aligned to the host genome (hg38) with bowtie2 (Presets: `--very-sensitive-local`) (*Langmead and Salzberg, 2012*), and the matched reads were discarded. The clean reads were assembled into contigs using MEGAHIT software (--k-list 21,33,55,77) (*Li et al., 2015*) with a filtering threshold of 500 bp. The ORFs were predicted using Prodigal software (*Hyatt et al., 2010*) and were translated into amino acid sequences. CD-HIT software (*Li et al., 2001*) was used to remove redundancy from the ORF predictions of each sample and mixed assembly with default parameyers, and to obtain non-redundant initial Unigenes. The clustering was performed with default parameters of 95% identity and 90% coverage, and the longest sequence was selected as the representative sequence. Bowtie2 software was used to align the clean reads of each sample to the non-redundant gene set (95% identity), and the abundance information of genes in each corresponding sample was calculated from the number of reads and the length of the genes.

## Taxa and functional annotation

The representative sequences of the redundant gene set were aligned with NCBI's NR database using DIAMOND software (*Buchfink et al., 2015*), and annotations with e < 1e−5 were selected to obtain proteins with the highest sequence similarity, thereby obtaining functional annotations and species annotations. Species present in only <10% of samples were excluded. The Valencia software (*France et al., 2020*) was used to perform CST assignment for each sample. The abundance of species is calculated using the sum of the gene abundance corresponding to the species. LEfSe (Linear discriminant analysis Effect Size) (*Segata et al., 2011*) was used to screen differentially abundant species by using MicrobiotaProcess R package (*Xu et al., 2023*). The first comparison uses the Kluskal–Wallis test, the second comparison uses the Wilcoxon test, and finally LDA is used to determine the difference, and the p value is corrected using False discovery rate (FDR). The calculation and visualization of species diversity and abundance are executed by the EasyMicroPlot R package (*Liu et al., 2021*) and EasyAmplicon pipline (*Liu et al., 2023*). Annotations were performed using the KO, COG, and CAZy databases to explore changes in enzyme family related to the microbiome. Differential functional analysis was conducted using the STAMP software (*Parks et al., 2014*).

## Construction of co-occurrence network

In network construction, we used more stringent species filtering criteria and only included species detected in 50% of samples in the analysis. The construction and random robustness comparison of fungal and bacterial–fungal co-occurrence networks were executed by the ggClusterNet R package (*Wen et al., 2022*). Spearman correlation analysis was used to calculate correlations, the cutoff value of the correlation coefficient was set to 0.6, and p values were corrected using FDR. The stability analysis of the network is implemented by the Robustness.Random.removal function of the ggClusterNet R package with the default parameters.

## Metabolite detection using LC and MS

A separation buffer (methanol/acetonitrile/water [2:2:1]) was used to extract metabolites from cervical swabs. Ten µl was taken from each sample and mixed to form a quality control sample (*Cai et al., 2015*), which was used to evaluate stability during the experiment. The metabolite signals in each sample were detected using LC and MS (ACQUITY UPLC I-Class plus, Waters). Progenesis QI v2.3 software was used for qualitative analysis of metabolites, with parameters of 5 ppm precursor tolerance, 10 ppm product tolerance, and 5% product ion threshold (Nonlinear Dynamics, Newcastle, UK). All output data were normalized using internal standards, and the results were displayed as peak values (test sample peak area/internal standard sample peak area). Compound identification was performed using human metabolome database (HMDB), Lipidmaps (V2.3), Metlin, EMDB, PMDB, and a self-built database based on mass-to-charge ratio (*m/z*), secondary fragmentation, and isotope distribution.

The screening of differential metabolites was performed using the OPLS-DA (orthogonal partial least squares discriminant analysis) method, with a VIP (Variable important in projection) value >1 for the first principal component and a p value <0.05 for *T*-test as the threshold. MetPA was used

for differential metabolic pathway analysis (*Xia and Wishart, 2010*). We used Spearman correlation analysis to evaluate the correlation between the microbial community and metabolites. The network was visualized using Cytoscape software (*Shannon et al., 2003*).

## Acknowledgements

The authors thank to OE Biotech Co, Ltd (Shanghai, China) for assistance with metagenomic sequencing. Funding This work was supported by the Guangdong Natural Science Foundation (2023A1515012940) and Postdoctoral Fellowship Program of CPSF (GZC20233216).

## Additional information

### Funding

| Funder | Grant reference number | Author |
|---|---|---|
| Natural Science Foundation of Guangdong Province | 2023A1515012940 | Xing Yang |
| Guangdong Basic and Applied Basic Research Foundation | 2023A1515110325 | Peigen Chen |
| Postdoctoral Fellowship Program of CPSF | GZC20233216 | Peigen Chen |

The funders had no role in study design, data collection, and interpretation, or the decision to submit the work for publication.

### Author contributions

Peigen Chen, Conceptualization, Data curation, Writing - original draft; Haicheng Chen, Conceptualization, Formal analysis, Validation; Ziyu Liu, Xinyi Pan, Data curation, Validation; Qianru Liu, Data curation, Validation, Visualization; Xing Yang, Conceptualization, Funding acquisition, Writing - review and editing

### Author ORCIDs

Peigen Chen http://orcid.org/0000-0002-0843-7739
Xing Yang http://orcid.org/0000-0002-2938-0268

### Ethics

The subjects were recruited at the Reproductive Medical Center of the Sixth Affiliated Hospital of Sun Yat-Sen University. The inclusion and exclusion criteria were previously described in a previous study (Yang et al 2022). All participants were fully informed, volunteered to participate in this study, and signed informed consent forms. All study processes were reviewed and approved by the Sixth Affiliated Hospital Ethics Committee of Sun Yat-Sen University (IRB no. 2019ZSLYEC-005S).

Reviewer #2 (Public Review): https://doi.org/10.7554/eLife.90363.4.sa1
Author response https://doi.org/10.7554/eLife.92562.sa2

## Additional files

### Supplementary files
• Supplementary file 1. 431 fungal species in all samples.
• MDAR checklist

### Data availability
The metagenome sequencing has been deposited in China National Center for Bioinformation (https://ngdc.cncb.ac.cn/) under reference number PRJCA016850.

The following dataset was generated:

| Author(s) | Year | Dataset title | Dataset URL | Database and Identifier |
|---|---|---|---|---|
| Chen P, Chen H, Liu Z, Pan X, Liu Q, Yang X | 2023 | Comprehensive analysis of bacterial-fungal interaction in Caesarean section scar diverticulum | https://ngdc.cncb.ac.cn/gsa-human/browse/HRA004716 | China National Center for Bioinformation, PRJCA016850 |

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
