## [Editor Report · eLife assessment]

This **important** study reports the fungal composition and its interaction with bacteria in the Caesarean section scar diverticulum. The data are **solid** and supportive of the conclusion. This work will be of interest to researchers and clinicians who work on women's health.

---

## [Referee Report · Reviewer #2 (Public Review)]

Summary:

Shotgun data have been analysed to obtain fungal and bacterial organisms abundance. Through their metabolic functions and through co-occurrence networks, a functional relationship between the two types of organisms can be inferred. By means of metabolomics, function-related metabolites are studied in order to deepen the fungus-bacteria synergy.

Strengths:

Data obtained in bacteria correlate with data from other authors.

The study of metabolic "interactions" between fungi and bacteria is quite new.

The inclusion of metabolomics data to support the results is a great contribution.

Weaknesses:

All my concerns have been clarified

---

## [Author Response]

The following is the authors’ response to the previous reviews.

Thank you once again for your patience and guidance through this revision process. I would like to add an important aspect to our previous discussion regarding the identification and impact of potential contaminants in our study.

In recent years, advanced tools such as SCRuB (recently published in Nature Biotechnology, DOI:10.1038/s41587-023-01696-w) and the widely-used tool decontam have been developed to address the issue of contaminants in metagenomic studies. These tools primarily operate based on sequence similarity, identifying potential contaminants by marking and removing those found in only a minority of samples or those that display patterns indicative of laboratory contamination.

As the reviewer rightly pointed out, contaminants are often rare species that appear in very few samples. Our study, focusing on high-abundance species in the vaginal microbiome, is less susceptible to the influences of such rare contaminants. This approach aligns with the methodology employed by leading research groups in the field, such as Professor Jacques Ravel's lab. Their decision not to use blank controls in several of their studies on the female reproductive tract microbiome likely stems from a similar understanding — that the impact of rare contaminants is minimal on the study's conclusions, especially when high-abundance species are the main focus.

We believe that the methodologies and tools currently available for contaminant identification and removal, while highly effective for their intended purpose, reinforce our decision to focus on high-abundance species. This focus minimizes the potential impact of rare contaminants on our study's conclusions. In light of this, our study's methodology remains robust and well-suited for achieving our research objectives.

In our revised manuscript, we will include a discussion of these points, further clarifying our approach and the rationale behind our methodological choices. We hope that this additional information will address the concerns raised and provide a clearer understanding of the context and reliability of our findings.

Thank you for considering these additional points. We look forward to your feedback on our revised manuscript.